# Response of Organic Fertilizer Application to Soil Microorganisms and Forage Biomass in Grass–Legume Mixtures

Huilin Yan [1,†], Xueli Zhou [1,2,†], Kaifu Zheng [1], Songsong Gu [1,3], Hao Yu [4], Kun Ma [1], Yangan Zhao [1], Yingcheng Wang [1], Hua Zheng [5], Hanjiang Liu [2], Dejun Shi [2], Guangxin Lu [1,*] and Ye Deng [3,*]

[1] College of Agriculture and Animal Husbandry, Qinghai University, Xining 810016, China
[2] Qinghai Province Grassland Improvement Experimental Station, Gonghe 813000, China
[3] CAS Key Laboratory of Environmental Biotechnology, Research Center for Eco-Environmental Sciences, Chinese Academy of Sciences, Beijing 100085, China
[4] College of Environmental Science and Engineering, Liaoning Technical University, Fuxin 123000, China
[5] State Key Laboratory of Urban and Regional Ecology, Research Center for Eco-Environmental Sciences, Chinese Academy of Sciences, Beijing 100085, China
[*] Correspondence: lugx74@163.com (G.L.); yedeng@rcees.ac.cn (Y.D.)
[†] These authors contributed equally to this work.

**Abstract:** It has been widely recognized that organic fertilizer (OF) application under monoculture and continuous cropping can change the microbial community and increase forage biomass in the Qinghai–Tibet Plateau. However, as a commonly used grassland planting pattern, the way in which grass–legume mixtures respond to OF application remains unclear. To clarify application effects of organic fertilizer in the grass–legume mixtures, we conducted a field experiment at the Qinghai–Tibet Plateau and collected the rhizospheric and bulk soils to reveal their microbial community by using high-throughput sequencing and molecular ecological networks. It was found that OF application changed the microbial community and increased the forage biomass under monoculture. However, in grass–legume mixtures, we found that OF application did not promote the increase of forage (Gramineae) biomass (Student $t$-test: $p > 0.05$). By analyzing both prokaryote and fungal communities, it was found that OF application had a greater impact on bulk soil microorganisms than on those of the rhizosphere in grass–legume mixtures. Co-occurrence network analysis showed that the rhizosphere and bulk soil networks of grass–legume mixtures were significantly more vulnerable under OF treatment (vulnerability of prokaryotes in grass: 0.1222; vulnerability of prokaryotes in legumes: 0.1730; fungal vulnerability in grass: 0.0116; fungal vulnerability in legumes: 0.0223) than non-OF treatment (vulnerability of prokaryotes in grass: 0.1015; vulnerability of prokaryotes in legumes: 0.1337; fungal vulnerability in grass: 0.0046; fungal vulnerability in legumes: 0.0126), which indicated that OF application did not provide favorable conditions for microbial interactions in grass–legume mixtures. In addition, structural equation modeling showed that OF application had some significant negative impacts on soil physicochemical properties and the robustness of the prokaryote community. The robustness of fungi had a significant negative ($p < 0.001$) impact on forage biomass, but OF application had no significant ($p > 0.05$) direct impact on the forage biomass, which indicated that the OF did not promote forage biomass in grass–legume mixtures. These results suggest that the application of organic fertilizer is unnecessary for grass–legume mixtures, because it does not promote the interactions between rhizospheric microbes and forage.

**Keywords:** organic fertilizer; grass–legume mixtures; Qinghai–Tibet Plateau; co-occurrence network; agriculture ecosystems; soil microorganisms





## 1. Introduction

The average altitude of the Qinghai–Tibet Plateau (QTP) is more than 4000 m. It is the highest plateau in the world and an ecologically fragile area [1,2]. Approximately 85% of the QTP is alpine grassland, most of which is utilized for grazing animal husbandry [3]. In

recent years, due to various activities, especially overgrazing, the alpine grasslands of the QTP have been seriously degraded. Year-round overgrazing leads to the reduction of soil organic matter and will eventually lead to desertification [4]. Therefore, the maintenance of ecosystem services contributes to the sustainable development of alpine grassland and the restoration of degraded grassland. Due to its unique alpine climate and harsh environment, the QTP is scarce in forage species, where Poaceae, Gramineae, and Cyperaceae are the main grass species. The establishment of artificial grassland can effectively alleviate grassland degradation. Artificial grassland is established to relieve the grazing pressure of natural grassland. The overgrazing of natural grassland reduces the degradation of grassland; contributes to grassland restoration; increases the grass biomass and vegetation coverage of alpine grassland; and, thus, enhances the carbon stock capacity; slows down the trend of global warming; protects species diversity and production function diversity; and increases soil fertility at the same time. Grass(Gramineae)–legume mixture planting is one of the cultivation methods of artificial grassland, which can both improve productivity and reduce the demand for nitrogen fertilizer [5]. where *Medicago sativa* L., 'Chuancao No.2', and *Elymus sibiricus* L. cv are generally used as forage varieties to establish artificial grassland [6].

Due to the scarcity of soil nutrients, agricultural management measures of fertilization have been used to improve the physical and chemical properties of soil, to promote soil microbial biomass and diversity, and to improve forage quality and yield in the QTP [7,8]. However, the long-term usage of chemical fertilizers not only leads to soil consolidation, but also to increased soil and atmospheric pollution, which seriously restricts green development [9]. In recent years, forage cultivation and breeding have increasingly adopted organic fertilizer (OF). OF is mainly composed of compounds that are naturally produced by the physical excretion or processing of artificially assisted crop straws or by-products; it improves soil health and gradually releases nutrients into the soil [10]. In the QTP, OF application is often used in the forage monoculture [2], but the application of top-dressing OF is used far less often for grass–legume mixtures. Therefore, little is known about the effect of OF application on forage yield in these latter systems and the mechanisms behind it.

Soil microbes give play to key role in maintaining soil health, forage diversity, and stability [11]. Rhizosphere microorganisms, as the second genome of plants, play an important role in the nutrient input and abiotic stress of forages. Rhizosphere-promoting bacteria (e.g., rhizobium) directly or indirectly promote plant growth through symbiotic relationships with host plants [12]. Previous studies have shown that OF application can not only change soil substrate availability, which in turn affects fungal diversity and community composition, but it also increases soil organic matter content, which improves soil nutrients and alters prokaryote communities [13,14]. Ling et al. [15] used PCR-DGGE technology to analyze the effect of continuous OF application on soil bacteria in watermelon continuous cropping, and found that the proper regulation of soil microorganisms was beneficial to plant health. A previous study that used molecular ecological networks to examine the relationship between soil microbes and soil under long-term organic fertilizer application showed that compared with bacteria, fungi are more sensitive to OF [16]. However, there have been few studies about the effects on soil diversity and structure and forage biomass in grass–legume mixtures receiving OF treatment.

Co-occurrence network can analyze the differences of microbial interactions in different habitats, and is a common method for microbial ecological analysis [17]. The distribution of the network model structures and linkages of microbial communities can provide strong predictions for ecosystem function and stability [18,19]. However, it is not clear how the application of OF changes the network structure of soil microbes in the rhizosphere and bulk soils of grass–legume mixtures. Structural equation modeling (SEM) is a method to establish, estimate and test the causal relationship model. It is an important tool for multivariate data analysis and is widely used in various scientific disciplines. Previous research has explored the relationship between environmental factors, microorganisms, and plant growth through structural equation modeling [5]. Our study used SEM to verify

the effect of organic fertilizer application on soil micro-organisms, and the relationship between the stability of soil microbial network and forage biomass.

Although there have been many studies on the effects of OF on soil microorganisms, its effects on soil microorganisms and forage biomass in grass–legume mixtures are still unclear. In order to elucidate the effect of OF application on soil physicochemical properties; microbial diversity and community structure; and the forage biomass of grass–legume mixtures in the QTP, we carried out mixed seeding experiments with and without organic fertilizers utilizing *Elymus sibiricus* L. cv. 'Chuancao No.2', *Elymus nutans* Griseb. cv. 'Aba', and *Medicago sativa* cv. 'Beilin 201' [2]. The molecular ecological network method and the effect of microbial community structure and its relationship with forage biomass on QTP in mixed was studied by SEM. To elucidate the response of organic fertilizer on soil microorganisms and the forage biomass of the grass–legume mixtures grassland, we hypothesized that (i) OF application would have a greater impact on bulk soil microorganisms than on rhizosphere organisms in grass–legume mixtures, (ii) after OF application, the microbial community of grass–legume mixtures would not be conducive to the growth of forage, and (iii) the stability of the rhizosphere and bulk soil networks in mixed sowing would be decreased by OF application.

## 2. Materials and Methods

### 2.1. Study Sites and Sampling

The experimental site was launched in Qinghai Grassland Improvement Experimental Station (99°35′ E, 37°05′ N) in June 2020 in the Qinghai–Tibet Plateau of China. The test station is located on the west bank of Qinghai Lake and is adjacent to Qinghai Lake in the east, which has obvious geographical advantages. The average altitude within the territory is 3270 m, the annual temperature is $-0.7$ °C, the annual precipitation is 1495.3 mm, the average annual precipitation is 368.1mm, the plateau climate characteristics are typical, the grassland types are diverse, and the site consists of alpine grassland.

There are 36 test plots with the size of $3 \times 5$ m; among them, 12 were OF or non-OF in monoculture, and 24 were OF or non-OF in grass–legume mixtures (12 were Beilin201 and Chuancao; 12 were Beilin and Aba). Organic fertilizer is provided by Resource College of Sichuan Agricultural University, the main raw materials are poultry manure and crop straw, adding biological starter culture, its content is inorganic substance content $\geq 45\%$, $N + P_2O_5 + K_2O \geq 5\%$, and pH value between 5.5–8.5. The treatment conditions were set as non-OF and 50 kg/acre of OF; no chemical fertilizer was added, and 3 replicates were established per treatment. The monoculture plots were sown with the gramineous grass *Elymus nutans* Griseb. cv. 'Aba' (Aba). The grass–legume mixtures, Beilin 201 (*Medicago sativa* cv. 'Beilin 201') was sown at 45 g/15 m$^2$ and Chuancao and Aba were sown at 22.5 g/m$^2$. Beilin 201 was mixed with Chuancao (*Elymus sibiricus* L. cv. 'Chuancao No.2') and Aba (*Elymus nutans* Griseb. cv. 'Aba') respectively.

Soil samples and forage biomass were collected on 13 September 2020 based on a random sampling method. During the forage growth period, 10 forage plants (grass only under monoculture and mixed sowing) were randomly selected from each plot; their aboveground and belowground biomass were measured; they were killed for 1 h at 105 °C and then dried for 48 h at 70 °C; and record the dry weight of dried grass.

A 50 cm × 50 cm quadrate was selected for sampling, and rhizosphere and bulk soil were collected. At the seedling stage of herbage growth, legume and grass forage were similar in root length and diameter, so we collected 15 cm of soil, including the roots of forage. After shaking off the large amounts of soil at the roots, the roots of all forage (legume and grass were separated for the grass–legume mixtures) were rinsed with phosphate buffered brine (PBS) 3 times. The suspensions were collected separately and centrifuged, and the supernatant was discarded. The supernatant (rhizosphere sample) was freeze-dried by vacuum drying pump (Boyikang Beijing Instrument Co., Ltd. FD-1C-50) and stored at $-80$ °C for DNA extraction. Soil temperature (MAT), soil moisture (SMC) and electrical conductivity (EC) were measured using a three parameter probe (Spectrum,

U.S. TDR 350). Under the monoculture planting pattern, the samples included 6 bulk soil samples of OF (BG), 5 bulk soil samples of non-OF (CKBG), 6 rhizosphere samples of OF (G), and 5 rhizosphere samples of non-OF (CKG). Under the grass–legume mixture planting pattern, the samples included 12 bulk soil samples of OF (BF), 12 bulk soil samples of non-OF (BCK), 9 rhizosphere samples of grass OF (MG), 9 rhizosphere samples of grass non-OF (CKMG), 10 rhizosphere samples of legume OF (ML), and 11 rhizosphere samples of legume non-OF (CKML). Each sample was a composite of five sub-samples of the same plant type from the same plot (Table S1).

### 2.2. DNA Extraction, Amplification, Sequencing, and Sequence Analysis

Total DNA was extracted in duplicate using the FastDNA™ SPIN kit (MoBio Laboratories, Carlsbad, CA, USA) following the manufacturer's instructions. DNA concentration above 20 ng/μL was measured by NanoDrop2000 spectrophotometer (Thermo), with A260/A280 values of 1.8~2.0.

Universal primers 515F (5′-GTGCCAGCMGCCGCGGTAA-3′) and 806R (5′-GGA-CTACHVGGGTWTCTAAT-3′) were used to amplify the 16S rRNA genes combined with self-designed barcodes to distinguish samples [20]; the polymerase chain reaction (PCR) amplification method followed a previously published protocol [5]. The ITS2 region fragment was amplified using the universal primer 5.8F-Fun (5′-AACTTTYRRCAAYGGATCWCT-3′) and ITS 4R-Fun (5′-AGCCTCCGCTTATTGATATGCTTAART-3′) [21]. The thermal cycle conditions of PCR amplification were 94 °C 3 min, 45 cycles 94 °C 20 s, 57 °C 25 s, 72 °C 45 s, 72 °C finally extended for 10 min. Positive PCR amplification was purified by 1% agarose gel electrophoresis and purified by kit (D2500-02, OMEGA BioTek). The purified am-plicons are quantified by NanoDrop2000 spectrophotometer. The concentration of all samples is above 20 ng/μL, and the ratio of A260/280 is 1.8~2.0. The samples were then mixed in equal proportions (150 ng) using a qubit fluorometer (Life technologies Holdings Pte Ltd., Singapore), The VAHTS™ nanoDNA library preparation kit for Illumina®(Vazyme Biotech Co., China Nanjing, China) was used and the library was constructed according to its reference instructions. The purified amplicons were gathered together and sequenced on the Hiseq sequencer (Illumina) at Magingene Biotechnology Co., LTD. (Guangzhou, China).

### 2.3. Sequence Data Preprocessing and Bioinformatics Approaches

The raw 16S rRNA gene and ITS2 fragment sequencing data were processed using an in-house pipeline (http://mem.rcees.ac.cn:8080 (accessed on 21 April 2022) integrated with various bioinformatics tools [22]. First, sequences were assigned to individual samples after detecting the barcodes (sample sorting). After removal of the barcode and primer sequences, the pair-ended sequences for 16S rRNA genes were merged using the Flash program [23]. For the ITS2 sequences, the forward and reverse primers were removed, leaving the target sequences [24]. The sequence is then passed through the ITSx program to remove ITS flank and non-fungal sequences. Then, Unoise3 algorithm (removing OTUs with abundance less than 8 sequences) was used to cluster the sequences into operational taxonomic units (OTUs), and the algorithm was applied to generate zOTUs(zero-radius OTUs) table. [25]. The taxonomic information annotations of the prokaryotes and fungi were performed based on RDP (training set No: 18 July 2020) and warcup databases V2 release in June 2016. Finally, 217,985 and 53,996 zOTUs were obtained for prokaryotes and fungi, respectively, before being used for downstream analysis.

### 2.4. Statistical Analysis

In this study, three measurements of alpha diversity, observed richness, Pielou evenness, and Simpson evenness, were calculated to assess the microbial communities in OF and non-OF. Richness was obtained by counting the observed number of species displayed in the zOTU tables. Pielou evenness represents the ratio of the actual Shannon index of a community to the maximum Shannon index that can be obtained from a community with the same species richness. Non-metric multidimensional scaling analysis (NMDS) was

used to analyze the β diversity of microbial communities in the bulk and rhizosphere soils for both OF and non-OF. Jaccard distance and dissimilarity tests were used to compare community structure between of and non-of groups based on MRPP, one-way analysis of variance (ANOSIM) and PERMANOVA. The above analysis was performed in the pipeline described earlier. LEfSe (linear discriminant analysis (LDA) effect size) analysis was used to further explore the markers of OF application in soil microbial communities, and then the differences of community abundance between groups were analyzed. SEM was built by IBM SPSS AMOS 26.0 Graphics (AMOS Development Corporation). X2/df is the fitting measure, expressed by approximate root mean square error (RMAESA) ($\leq$0.1, indicating relatively close), degree of fit index (GFI) ($\geq$0.90, indicating relatively accurate fit) [26], and comparative fit index (CFI) ($\geq$0.90, indicating relatively accurate fit) [27]. Rhizosphere and bulk soil prokaryote and fungal community composition, network community, forage biomass (i.e., above and belowground biomass) and soil physicochemical properties (i.e., TN, TOC, SMC, EC) were the first component representation of the principal component analysis (PCA).

*2.5. Source Tracker Analysis*

Source tracker [28] is an effective tool for microbial source tracking and analysis. After removing all samples with less than abundance 1% from the zOTU tables, the proportion of main sources of bacteria and fungi in the rhizosphere was estimated in monoculture and grass–legume mixtures after OF application, and the percentage was the resulting average.

*2.6. Stochastic Matrix Theory Based Molecular Ecology Networks and Analysis*

To uncover the effects OF of on soil microbial interactions, we construct molecular ecological networks using publicly pipeline (http://mem.rcees.ac.cn:8081 (accessed on 22 April 2022) [29]. In this study, based on stochastic matrix theory (RMT), pMENs were constructed from the OTU relative abundance of bacteria and fungi in the rhizosphere and bulk soils, and the topology of the network was identified by this method (i.e., network nodes, links, average degree (avgK), average clustering coefficient (avgCC), etc.), and 100 random networks were generated based on the Maslov–Sneppen program. At the same time, in order to prove whether and how the application of OF affected the stability of the network, the robustness and vulnerability of the networks were calculated. Robustness is defined as the proportion of species remaining in the network after the random removal of 50% of nodes [17,30]. Vulnerability represents the relative contribution of computed nodes to global efficiency [31]. Gephi (v0.9.2; https://gephi.org/ (accessed on 15 May 2022) was used to visualize the network.

## 3. Results

*3.1. Effects of Organic Fertilization on Physical and Chemical Properties of Soil, and Forage Biomass of Grass–Legume Mixtures*

ANOVA analysis showed that soil carbon content had significant difference ($p < 0.05$) in soil carbon and total nitrogen concentration between the OF and non-OF soils of grass–legume mixture treatments (Table S2). There were no significant differences among the rhizosphere soils. In the rhizosphere and bulk soils, the TN, NO$_3$-N, NH$_4$-N, and TOC of the OF treatment were higher than those of the non-OF. In bulk soil, TN increased by 9.41%, and TOC increased by 17.01% after OF application. At the same time, the aboveground biomass (i.e., plant fresh weight, dry weight) and belowground biomass of the grass were also analyzed. The results display that there was no significant difference between the aboveground biomass and belowground biomass of the grasses in grass–legume mixtures with OF (MG) and that of the grass in the monoculture with non-OF (CKG). Compared with non-OF treatment (CKMG), OF application (MG) did not significantly increase the aboveground and belowground biomasses of forage in the grass–legume mixtures, as is shown in Table S2 and Figure 1.

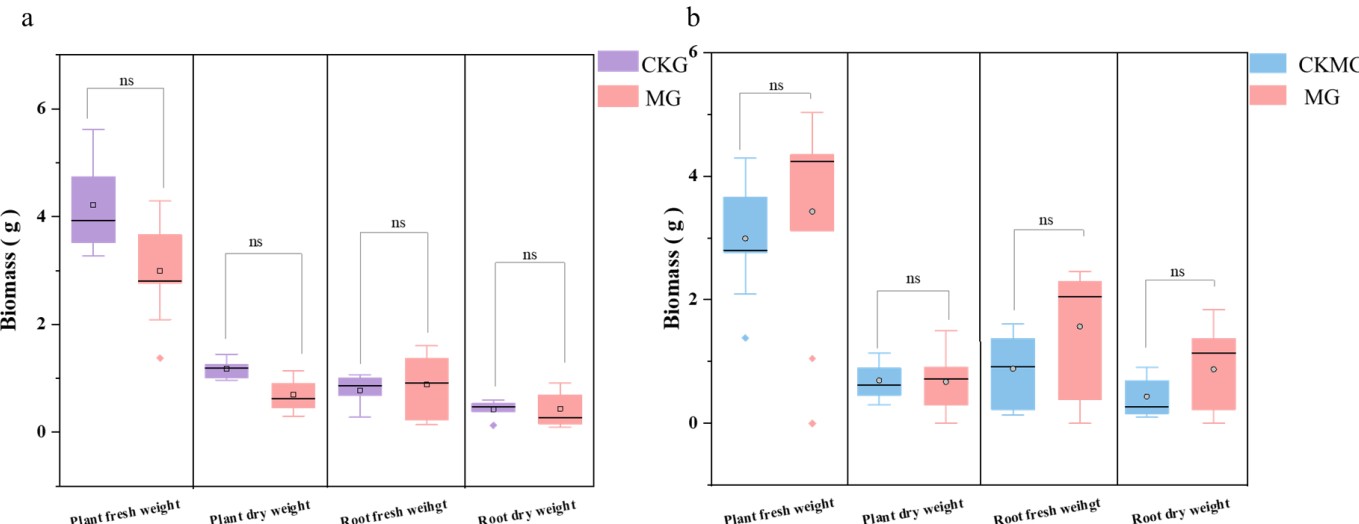

**Figure 1.** Comparison of forage biomass under different planting patterns. (**a**) Comparison of unfertilized monoculture grass biomass and organic fertilized mixed sowing grass; (**b**) Comparison of grass biomass in mixed sowing with and without organic fertilizer; ns means that there is no significant difference between the groups. CKG represents the grass that did not apply organic fertilizer in monoculture; MG represents the grass with mixed sowing and fertilization. CKMG represents the grass which was mixed without organic fertilizer.

### 3.2. Effect of Organic Fertilization on Microbial Community Diversity, Structure, and Composition

The 16S rRNA genes and ITS genes of 85 samples were amplified to detect prokaryotic and fungal communities, respectively. The 16S rRNA gene sequences were clustered into 217,985 zOTUs, and then resampled to 40,175 sequences of each sample for further statistical analysis. ITS gene sequences were clustered into 53,996 zOTUs, and then resampled to 28,918 sequences in each sample for further statistical analysis.

Observed richness, Pielou evenness, and Simpson evenness were calculated to evaluate $\alpha$ diversity of the prokaryote and fungal communities with OF and non-OF treatment. In the monoculture, the observed richness and Simpson evenness of the prokaryote communities in the rhizosphere and bulk soils showed significant differences ($p < 0.05$) between OF and non-OF treatments. There were also no dramatic Difference in the observed richness and Pielou evenness of the fungal communities between the OF and non-OF in bulk soil. In the grass–legume mixtures, the Pielou evenness and Simpson evenness of the prokaryote and fungal communities in the rhizosphere and bulk soils declared no significant ($p > 0.05$) difference between OF and non-OF treatments (Figure S1). The composition and structure of prokaryotes and bacteria were analyzed. On the basis of the NMDS bottom on Jaccard distances, the prokaryote communities in the OF and non-OF could not be separated, while the fungal communities in the OF were partially separated (Figure S2).

In the monoculture, dissimilarity tests, including MRPPs, ANOSIM and PERMANOVA, confirmed that there were significant differences in the rhizosphere prokaryote community after OF application compared with that of the non-OF (MRPP, $p = 0.025$; ANOSIM, $p = 0.041$; PERMANOVA, $p = 0.028$). While there were differences in the bulk soil prokaryote communities, they were not significant ($p > 0.05$). There was no significant difference in the rhizosphere and bulk soil fungal communities between the OF and non-OF. In the grass–legume mixtures, compared with the non-OF, there was no significant difference in the rhizosphere prokaryote community with OF, but there was a significant difference in the bulk soil prokaryote community (MRPP, $p = 0.01$; ANOSIM, $p = 0.026$; PERMANOVA, $p = 0.015$). In contrast, there were significant differences in rhizosphere fungal communities with OF (MRPP, $p = 0.03$; ANOSIM, $p = 0.037$; PERMANOVA, $p = 0.026$), as well as bulk soil fungal communities (MRPP, $p = 0.001$; ANOSIM, $p = 0.001$; PERMANOVA, $p = 0.001$). The dissimilarity test was conducted for the rhizospheres of the legumes and grass in the

grass–legume mixtures. It was found that, while there no distinct difference in the fungal communities of the non-OF treatment, there were significant differences in the rhizosphere fungal communities of the two forage species with OF (Table 1).

**Table 1.** Dissimilarity test by MRPP, ANOSIM, and PERMANOVA based on Jaccard dissimilarity between planting systems. In the monoculture, G represents rhizosphere samples of OF; CKG represents rhizosphere samples of non-OF; BG represents bulk soil samples of OF; CKBG represents bulk soil samples of non-OF. In the grass–legume mixture, MG represents rhizosphere samples of grass OF; CKMG represents rhizosphere samples of grass non-OF; ML represents rhizosphere samples of legume OF; CKML represents rhizosphere samples of legume non-OF; BF represents bulk soils samples of OF; BCK represents bulk soil samples of non-OF.

| Community | Area | Planting Pattern | Group | MRPP | | ANOSIM | | PERMANOVA | |
|---|---|---|---|---|---|---|---|---|---|
| | | | | Delta | $p$ | r | $p$ | Pesudo-F | $p$ |
| 16S rRNA gene | Rhizosphere | Monoculture | G vs. CKG | 0.7194 | 0.025 | 0.288 | 0.041 | 2.2664 | 0.028 |
| | | | MG vs. CKMG | 0.6453 | 0.493 | −0.0034 | 0.41 | 0.9052 | 0.607 |
| | | | ML vs. CKML | 0.6563 | 0.567 | −0.0232 | 0.569 | 1.0384 | 0.236 |
| | | Grass-legume mixture | MG vs. ML | 0.6588 | 0.34 | 0.0400 | 0.183 | 0.9679 | 0.401 |
| | | | CKMG vs. CKML | 0.6440 | 0.258 | 0.0078 | 0.31 | 1.0504 | 0.259 |
| | Bulk soil | Monoculture | BG vs. CKBG | 0.8167 | 0.129 | 0.2533 | 0.073 | 1.7621 | 0.112 |
| | | Grass-legume mixture | BF vs. BCK | 0.5278 | 0.01 | 0.0712 | 0.026 | 1.1398 | 0.015 |
| ITS gene | Rhizphere | Monoculture | G vs. CKG | 0.7660 | 0.79 | −0.0666 | 0.734 | 0.9781 | 0.786 |
| | | | MG vs. CKMG | 0.7587 | 0.094 | 0.0545 | 0.139 | 1.0905 | 0.095 |
| | | | ML vs. CKML | 0.7640 | 0.648 | −0.0307 | 0.762 | 0.9800 | 0.452 |
| | | Grass-legume mixture | MG vs. ML | 0.7418 | 0.03 | 0.1390 | 0.037 | 1.2312 | 0.026 |
| | | | CKMG vs. CKML | 0.7406 | 0.655 | −0.0331 | 0.692 | 0.9629 | 0.626 |
| | Bulk soil | Monoculture | BG vs. CKBG | 0.7800 | 0.973 | −0.136 | 0.935 | 0.9437 | 0.958 |
| | | Grass-legume mixture | BF vs. BCK | 0.7626 | 0.001 | 0.3685 | 0.001 | 1.6687 | 0.001 |

In the grass–legume mixtures, all ITS zOTUs were classified into 489 genera, belonging to six phyla, and the top 10 most abundant OTUs ($\geq 1.0\%$ relative abundance) of the rhizosphere and bulk soil are shown in Figure S3a. All 16S rRNA zOTUs were classified into 619 genera, belonging to 45 phyla, and the top 10 most abundant OTUs ($\geq 1.0\%$ relative abundance) of the rhizosphere and bulk soil are shown in Figure S3b. The bulk and rhizosphere soil fungal communities were dominated by *Ascomycota*, *Basidiomycota*, *Zygomycota*, *Glomeromycota*, and *Chytridiomycota* at the phylum level; in addition, compared with non-OF, the relative abundance of *Ascomycota* in the rhizosphere and bulk soils increased under OF treatment, while Basidiomycota decreased. At the genus level, the total soil fungal communities were mainly composed of *Gibberella*, *Atradidymella*, *Podospora*, *Fusarium*, *Didymella*, *Plectosphaerella*, *Preussia*, *Leucosporidium*, and *Thielavia*. The rhizosphere soil prokaryotic communities were dominated by Proteobacteria, Actinobacteriota, and Firmicutes, but, in the bulk soil, Proteobacteria, Actinobacteriota, and Acidobacteriota were the dominant phyla. At the genus level, the soil prokaryotic communities were mainly composed of *Exiguobacterium*, *Pseudarthrobacter*, and *Pseudomonas*, with the highest relative

abundance of *Exiguobacterium* found in the rhizosphere soil, while the relative abundance of *Exiguobacterium* was significantly ($p < 0.05$) lower in the bulk soil than in the rhizosphere soil. Additionally, the relative abundances of some genera in the rhizosphere soils and bulk soil were altered by OF application (Figure S3).

LEfSe analysis revealed significant differences between the MG and ML (linear discriminant analysis (LDA) > 2), and identified high-dimensional biomarkers in the OF treatments. LDA analysis of the fungal communities detected 21 (MG = 18, ML = 3) biometrically significant biomarkers. At the class level, the higher score biomarkers of the MG belonged to *Aganicaceae*, *Leucoagaricus*, *Unclassified*, *Entrophosporaceae*, *Entorphospora*, *Entrophosporaneradensis*, *Penicillium*, *Xylariaceae*, *Gliocephalotrichum*, *Gliocephalotrichunbacillisporum*, *Hypocrealixli*, *Crgytoloccuspodzolicus*, *Penicilliumerythromellis*, and *Xylaria*. The higher score biomarkers of ML belonged to *Fusariumacuminatum*, *Fusariumtorulosum*, and *Fusariumlateritium* (Figures 2a and S4a). LEfSe analysis showed that, in the bulk soil fungal community, *Fusariumtricinutum*, *Phomaglomerata*, and *Microbotryomycete* were significantly different under the OF treatment compared with non-OF treatment. In the bulk soil prokaryote community, *Pseudomonasfrederiksbergenis* and *Bacilluscirculans* were significantly altered after OF treatment compared with non-OF treatment (Figures 2b and S4b).

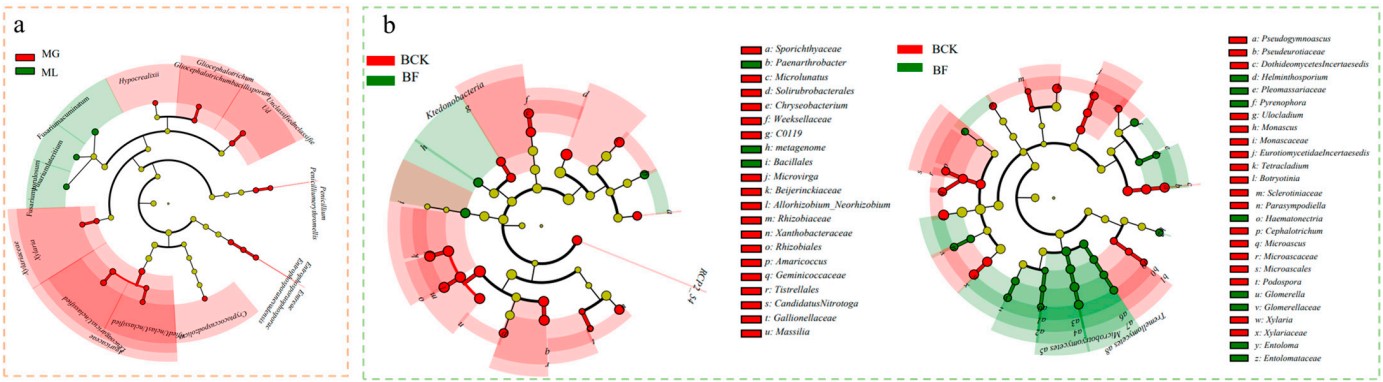

**Figure 2.** Differentiated taxa between groups and cladogram based on analyses results of LEfSe (with LDA score set as 2). (**a**) Comparison of grass (MG) and legume (ML) in rhizosphere fungal communities. (**b**) Comparison of bulk soil (BCK vs. BF) bacterial and fungal communities. From the inner circle to the outer circle of the cladogram, the yellow nodes indicate that there is no significant difference between the groups, while the other color nodes indicate that the group has a high abundance within the group and a significant difference between the groups.

*3.3. Source Tracker Analyzes the Main Source of Rhizosphere Prokaryote and Fungal Communities after Application of Organic Fertilizer*

After the application of OF, the monoculture rhizosphere microbial community changed significantly, while the rhizosphere microbial community of the grass–legume mixtures did not. Source tracker was used to analyze the main sources of the differences (Figure 3). In the monoculture, with no application of OF, 77.55% of the rhizosphere prokaryote community (CKG) was derived from the bulk soil prokaryote community, while with the application of OF, 81.84% of the rhizosphere prokaryote community (G) was derived from the bulk soil prokaryote community. In the rhizosphere fungal community, the percentages were 81.84% (CKG) and 77.58% (G) derived from the bulk soil fungal community. In the grass–legume mixtures, 58.02% of the rhizosphere prokaryote community (CKMG) of the grass was derived from the legume rhizosphere prokaryote community (CKML), and only 24.89% from the bulk soil prokaryote community (BCK) in the non-OF treatment. Under OF application, 58.50% of the rhizosphere prokaryote community (MG) of the grass was derived from the legume rhizosphere community (ML), and only 26.93% came from the bulk soil prokaryote community. Legume rhizosphere fungi (CKML) were the source of 51.88% of the grass rhizosphere fungal community (CKMG) without OF treatment. With the application of

OF, 48.42% of the grass rhizosphere fungal community (MG) was derived from the legume rhizosphere fungal community (ML).

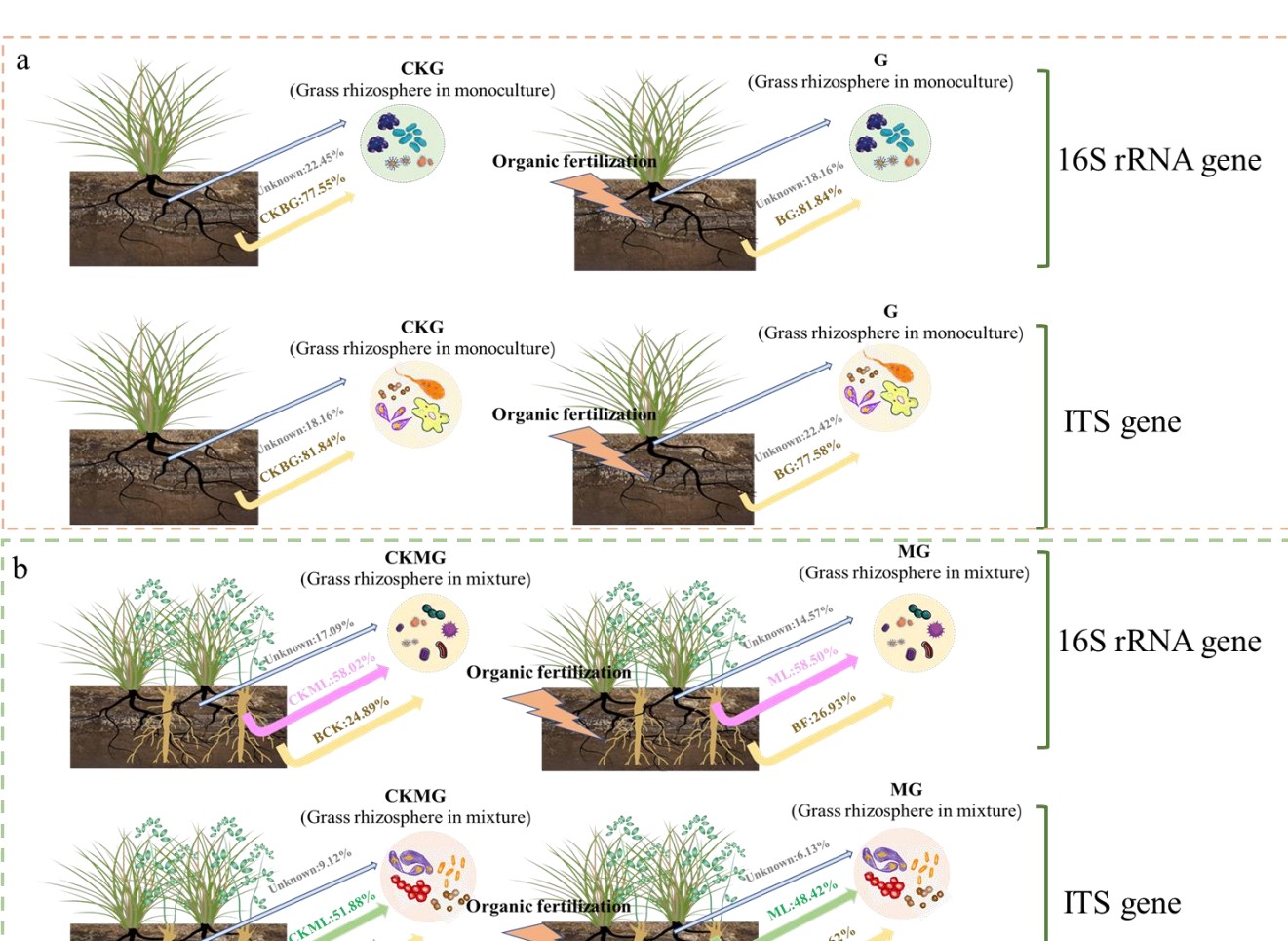

**Figure 3.** Source tracker analysis. Source tracker analysis proportion of rhizosphere microorganisms in monoculture and grass–legume mixtures. (**a**) Monoculture. (**b**) Grass–legume mixtures.

### 3.4. Molecular Ecological Network Analysis on Soil Communities

The interaction between OF and non-OF microbial communities was revealed by the analysis Molecular ecological networks (MENs), whose topological properties are shown in Figure 4 and Table 2. Average connectivity is used to assess network complexity. In the fungal networks and rhizosphere soils, the scale of non-OF networks was more complex than that of OF networks (avg K: CKMG of 3.023 > MG of 2.043; CKML of 3.022 > ML of 2.069), while the bulk soil showed the opposite trend of rhizosphere (avgK: BF of 3.134 > BCK of 2.855). The average path lengths in the OF and non-OF rhizosphere soil of the grass were MG (0.033) and CKMG (0.037), respectively, and in the legumes were ML (0.023) and CKML (0.034), respectively. The bulk soil in the OF and non-OF were 1.842 (BF) and 2.211 (BCK), respectively. In the prokaryote networks and rhizosphere soils, the network structure was more complex with OF than with non-OF (avgK: MG of 3.869 > CKMG of 3.663; ML of 4.329 > CKML of 4.292), and the bulk soil, again, showed the opposite trend of rhizosphere (avgK: BCK of 2.211 > BF of 1.842). The average path lengths in the OF and non-OF rhizosphere soils of the grass were 7.16 (MG) and 6.231 (CKMG), respectively, and in the legumes were 4.394 (ML) and 4.694 (CKML), respectively. The bulk soils in the OF and non-OF were 16.612 (BF) and 14.309 (BCK), respectively (Table 2), and these values

were very close to the logarithm of the total size of the network, which is significantly unlike from other networks, thus showing typical small-world network characteristics. These results show that all nodes were highly interconnected in the network. The differences in topological properties between the empirical networks and their corresponding random networks were compared for modularization analysis. Finally, in the fungal networks, the M for the OF and non-OF in the rhizosphere soils of the grass were 0.873 (MG) and 0.808 (CKMG), in the legumes were 0.963 (ML) and 0.809 (CKML), and the bulk soils for the OF and non-OF were 0.811 (BF) and 0.801 (BCK), respectively. In the prokaryote networks, the modularity values (M) for OF and non-OF in the rhizosphere soils of grass were 0.788 (MG) and 0.793 (CKMG), in legumes were 0.644 (ML) and 0.626 (CKML), and the bulk soils for OF and non-OF were 0.959 (BF) and 0.924 (BCK), respectively. These values were all higher than the M values of the corresponding random networks, thus indicating that the MENs constructed had modular structures. In addition, the results of the constructed random network showed that the network indicators (e.g., average clustering coefficient, average path length, and modularity) in the fungal network and the prokaryotic network are different between OF and non-OF. (Figure 2 and Table 2). These results proved that, in the rhizosphere soil prokaryote community, the application of OF reduced the network scale of grass, but increased the network scale of legumes. However, the network scales of rhizosphere soil fungi in both grass and legumes were reduced by OF application. The network scales of prokaryote and fungi in bulk soil were reduced by OF application.

In addition, we also calculated the robustness and vulnerability of each network, which represent network stability. In grass–legume mixtures, in the fungal community, the network robustness in MG (average $0.133 \pm 0.124$) was lower than in CKMG (average $0.183 \pm 0.166$), and ML (average $0.215 \pm 0.142$) was lower than in CKML (average $0.234 \pm 0.110$). In the bulk soil, BF (average $0.216 \pm 0.129$) was higher than in BCK (average $0.211 \pm 0.105$), but not significantly. In the prokaryote community, the robustness was significantly higher without fertilization than that under the application of OF, the MG (average $0.429 \pm 0.019$) was lower than the CKMG (average $0.440 \pm 0.018$), and the ML (average $0.423 \pm 0.019$) was lower than the CKML (average $0.446 \pm 0.016$). In the bulk soil, the BF (average $0.431 \pm 0.020$) was higher than the BCK (average $0.413 \pm 0.021$), which indicated that the network under the application of OF did not have higher stability (Figure 4b). The network vulnerability of the soil prokaryote and fungal communities was significantly increased after the application of OF in grass–legume mixtures. In the fungal community, the MG (0.1222) was higher than the CKMG (0.1015), and the ML (0.1730) was higher than the CKML (0.1337). In the bulk soil, the BF (0.248) was higher than the BCK (0.0527). In the prokaryote community, the MG (0.0116) was higher than the CKMG (0.0046), and the ML (0.0223) was higher than the CKML (0.0126). In the bulk soil, the BF (0.0697) was higher than the BCK (0.0104). Together, this indicated that the stability of soil prokaryote and fungal communities in bulk and rhizosphere soils decreased after the application of OF (Figure 4a).

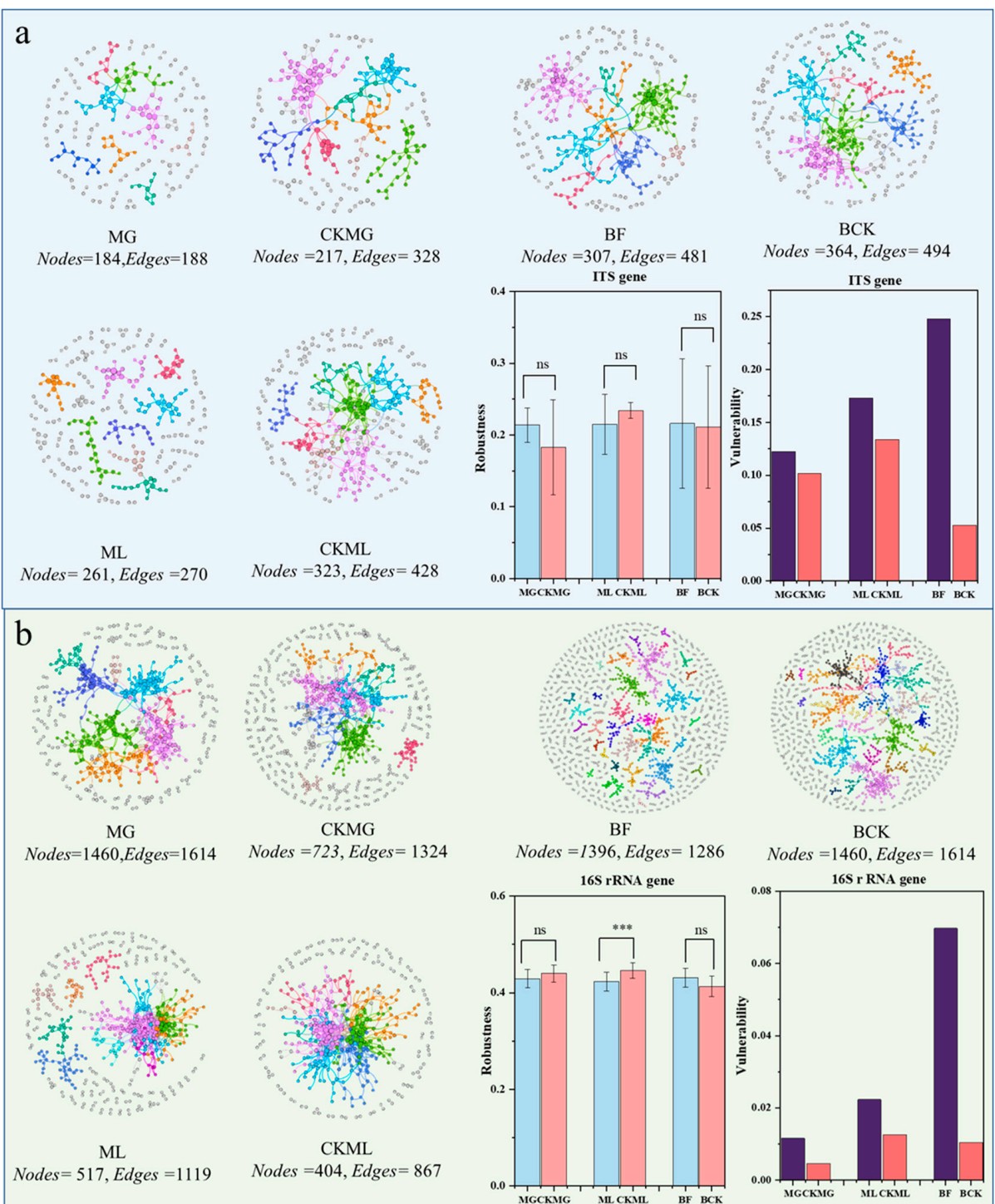

**Figure 4.** Networks of microbial communities in soils with or without OF amendment. Each node represents an OTU. The links between nodes show strong and significant correlation ($p < 0.01$). Modules are represented in different colors, and nodes in modules with less than 10 members are represented in gray. Visualization of the stability (robustness and vulnerability) of the rhizosphere soil microbial network in grass–legume mixtures. (**a**) Fungal community. (**b**) Prokaryote community. In the grass–legume mixtures, MG represents rhizosphere samples of grass OF; CKMG represents rhizosphere samples of grass non-OF; ML represents rhizosphere samples of legume OF; CKML-represents rhizosphere samples of legume non-OF; BF represents bulk soils samples of OF; BCK represents bulk soil samples of non-OF. *** represent significance at $p < 0.001$, ns represents no significance, respectively.

**Table 2.** Properties of empirical and randomized molecular ecology networks (MENs) for prokaryotic and fungal communities under OF and non-OF treatments. Randomized networks were performed by rewiring all the nodes and links corresponding to empirical networks 100 times.

| Group | Molecular Ecological Network | | | | | | | | | | | Random Network | | |
|---|---|---|---|---|---|---|---|---|---|---|---|---|---|---|
| | Similarity Threshold | Nodes | Links | Average Degree (avgK) | Average Clustering Coefficient (avgCC) | Average Path Distance (GD) | Centralization of Degree (CD) | Density (D) | Transitivity (Trans) | Modularity | $R^2$ | Average Clustering Coefficient (avgCC) | Average Path Distance (GD) | Modularity |
| MG | 0.970 | 488 | 944 | 3.869 | 0.293 [a] | 7.16 [b] | 0.025 | 0.008 | 0.402 | 0.788 [c] | 0.85 | 0.012 ± 0.003 | 4.324 ± 0.045 | 0.517 ± 0.006 |
| CKMG | 0.970 | 723 | 1324 | 3.663 | 0.259 [a] | 6.231 [b] | 0.021 | 0.005 | 0.449 | 0.794 [c] | 0.838 | 0.008 ± 0.002 | 4.564 ± 0.040 | 0.542 ± 0.004 |
| ML | 0.960 | 517 | 1119 | 4.329 | 0.259 [a] | 4.394 [b] | 0.046 | 0.008 | 0.445 | 0.645 [c] | 0.923 | 0.027 ± 0.004 | 3.869 ± 0.043 | 0.462 ± 0.005 |
| CKML | 0.960 | 404 | 867 | 4.292 | 0.217 [a] | 4.694 [b] | 0.049 | 0.011 | 0.364 | 0.626 [c] | 0.895 | 0.025 ± 0.005 | 3.839 ± 0.047 | 0.465 ± 0.006 |
| BF | 0.890 | 1396 | 1286 | 1.842 | 0.104 [a] | 16.612 [b] | 0.007 | 0.001 | 0.265 | 0.959 [c] | 0.93 | 0.001 ± 0.001 | 9.374 ± 0.332 | 0.899 ± 0.004 |
| BCK | 0.890 | 1460 | 1614 | 2.211 | 0.128 [a] | 14.309 [b] | 0.011 | 0.002 | 0.294 | 0.925 [c] | 0.947 | 0.002 ± 0.001 | 7.309 ± 0.119 | 0.806 ± 0.004 |
| MG | 0.890 | 184 | 188 | 2.043 | 0.149 [a] | 4.924 [b] | 0.033 | 0.011 | 0.313 | 0.873 [c] | 0.85 | 0.007 ± 0.006 | 6.188 ± 0.408 | 0.782 ± 0.011 |
| CKMG | 0.890 | 217 | 328 | 3.023 | 0.227 [a] | 8.712 [b] | 0.037 | 0.014 | 0.382 | 0.809 [c] | 0.806 | 0.015 ± 0.007 | 4.649 ± 0.102 | 0.601 ± 0.010 |
| ML | 0.860 | 261 | 270 | 2.069 | 0.188 [a] | 3.577 [b] | 0.023 | 0.008 | 0.381 | 0.937 [c] | 0.852 | 0.005 ± 0.004 | 7.173 ± 0.401 | 0.802 ± 0.009 |
| CKML | 0.860 | 323 | 488 | 3.022 | 0.26 [a] | 6.732 [b] | 0.034 | 0.009 | 0.323 | 0.8 [c] | 0.87 | 0.011 ± 0.004 | 4.925 ± 0.073 | 0.616 ± 0.007 |
| BF | 0.830 | 307 | 481 | 3.134 | 0.235 [a] | 10.505 [b] | 0.059 | 0.01 | 0.376 | 0.811 [c] | 0.864 | 0.013 ± 0.005 | 4.575 ± 0.076 | 0.594 ± 0.007 |
| BN | 0.830 | 346 | 494 | 2.855 | 0.205 [a] | 6.815 [b] | 0.035 | 0.008 | 0.296 | 0.802 [c] | 0.865 | 0.009 ± 0.004 | 4.972 ± 0.094 | 0.640 ± 0.007 |

[a]. Significant difference in avgCC between empirical and randomized networks based on Student's *t*-test; [b]. Significant difference in GD between empirical and randomized networks based on Student's *t*-test; [c]. Significant difference in M between empirical and randomized networks based on Student's *t*-test. The gray shading is the fungal network attributes.

### 3.5. Structural Equation Model Analysis of Relationship between Soil Microbial Community and Forage Biomass under Organic Fertilizer Treatment

SEM analysis further explored the relationship between soil physicochemical properties, microbial community characteristics, network stability, and forage biomass under OF treatment. The evaluation results showed that the standardization coefficient did not exceed or approach 1, the chi-square was 2.242, and the variance evaluation results did not show negative values. The GFI was 0.987, the CFI was 1.000, all of which indicated that the overall model fitted well. Our model explained 98% of the environmental factors (TN, $NO_3$-N, $NH_4$-N, TOC), 50% of the bulk soil prokaryote community, 93.9% of the forage biomass (plant fresh weight and plant dry weight), 61% of the rhizosphere prokaryote community, and 52% of both the rhizosphere and bulk soil fungal communities (the first coordinate of PCA) (Figure 5). The results showed that the application of OF had a significant negative correlation with environmental factors (−0.358) and a significant negative correlation with the stability of the prokaryote network (−0.934), but had no effect on forage biomass. The application of organic fertilizer had no effect on the rhizosphere and bulk soil fungal communities. In addition, the stability of the prokaryote network was significantly positively correlated with forage biomass (0.843), while the stability of the fungal network was significantly negatively correlated with forage biomass (−0.702). Comprehensive analysis showed that the application of OF changed the physical and chemical properties of soil but had no direct effect on the rhizosphere soil prokaryote and fungal communities, and had no direct promotive effect on the forage biomass under grass-legume mixtures.

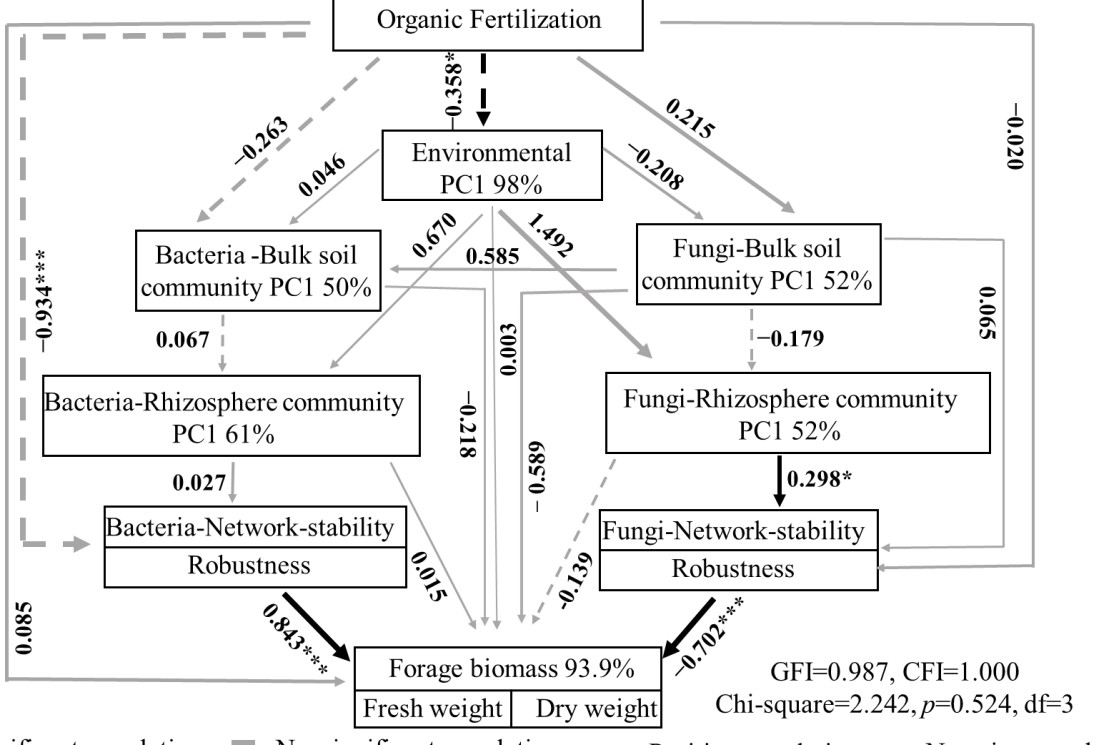

**Figure 5.** Model and estimate results of relationship between organic fertilizer application, microbial diversity, network attribute structure, and forage biomass. Black indicates significant correlation between potential variables, gray indicates non-significant correlation. Dashed arrows indicate negative correlation, while solid arrows indicate a positive correlation. The wider the arrow width, the greater the absolute value of the path coefficient. *** and * represent significance at $p < 0.001$ and $p < 0.01$, respectively.

## 4. Discussion

Bacteria and fungi play important roles in agro-ecosystems as two major components of the soil microbial community, and soil microbes are very sensitive to fertilization treatments [32,33]. However, less attention has noticed this effect of organic fertilizer application on the microbial community of grass–legume mixtures in the Qinghai–Tibet Plateau. Taking the area around Qinghai Lake on the QTP as the research site, the physicochemical properties, prokaryote and fungal community compositions, network analysis, and the relationship between OF application and forage biomass in grass–legume mixtures were studied. Our findings in the current study, that OF application to a monoculture altered microbial communities and increased forage biomass, were consistent with our previous work [2]. For the planting patterns of grass–legume mixtures, our results showed that the application of OF significantly changed soil properties, that is, key physicochemical factors were changed, such as TN and TOC concentrations, thereby affecting soil microbial diversity and community structure. However, there was no direct relationship between the microbial changes and forage biomass (Figure 1, Table S2). The microbial diversity of treated rhizosphere soil samples was higher in grass than without OF treatment, while the opposite was true for legumes (Figure S1b). The microbial diversity in bulk soil samples after treatment was lower than that without treatment. In the prokaryote communities, OF treatments increased network scale (more edges and nodes), while in the fungal communities, it decreased network scale. At the same time, OF did not improve the network stability (robustness and vulnerability) of prokaryote and fungal communities (Figure 4). SEM showed that the application of OF in prokaryote and fungal communities had a significant negative correlation between the application of OF and the overall soil physicochemical properties, and the application of OF had no significant positive correlation with forage biomass. The application of OF had a significant negative correlation with the stability of the prokaryote network structure, and while it also had a negative correlation with the stability of the fungal network, it was not significant. The stability of the prokaryote network had a significant positive correlation with forage biomass, while the stability of the fungal network had a significant negative correlation (Figure 5). These results confirmed the hypothesis that the application of OF altered soil microbial diversity and altered network structure.

Several studies have shown that the application of OF can significantly alter soil physicochemical properties [34,35], and our results are consistent with this (Table S2). Studies have shown that OF application in the Qinghai –Tibet Plateau can increase the plant height and aboveground biomass of grass forages, and have also shown that the application of organic fertilizers can promote plant growth [2,36]. However, in our study, we found that the application of OF did not increase the forage biomass for the planting patterns of grass–legume mixtures in the Qinghai–Tibet Plateau (Figure 1). Since the planting time of the experimental plots was slightly later than that of the Qinghai–Tibet Plateau, in addition to the harsh environment of the Qinghai–Tibet Plateau, with extreme weather such as hail and snow [37], grass grows slowly during the seedling stage. Previous studies used structural equation modeling to demonstrate that the increase in forage yield was directly related to microorganisms, but was indirectly or not related to soil physicochemical properties for the planting patterns of grass–legume mixtures in the Qinghai–Tibet Plateau [5]. Previous studies have shown that, after co-sowing black oats and clover, the nitrogen source of black oats was mainly derived from the clover, and the co-sowing of black oats and clover fully benefited from symbiotic $N_2$ fixation [38]. In our results, we found that, in the monoculture, the rhizosphere prokaryote and fungal communities of grass after OF application were mainly derived from the bulk soil prokaryote and fungal communities. Meanwhile, the rhizosphere prokaryote and fungal communities of the grass in grass–legume mixtures after OF application were mainly derived from the rhizosphere prokaryote and fungal communities of the legume. Previous studies have found that the fungal community in alpine regions is more sensitive to external disturbances than the prokaryote community, and the application of OF can lead to an increase in nitrogen, which changes microbial

community structure [2,39]. Our source tracker results showed that the rhizosphere grass fungal and prokaryote communities were mainly derived from the rhizosphere of the legumes in grass–legume mixtures (Figure 3). In summary, the results found that the forage biomass did not increase significantly, which may have been due to the presence of extreme weather and the fact that the legume symbionts supply enough nitrogen for the system, while the application of organic fertilizer created a nitrogen redundancy.

Soil microorganisms are very sensitive to environmental disturbances, and their diversity and community structure have important effects on soil physicochemical properties and soil ecological functions [40]. Studies have shown that the application of organic fertilizer can significantly change the Shannon and Simpson indices of microorganisms, and increase the functional diversity [41]. Our results showed that the microbial diversity of the bulk soil increased after the application of OF. Among the rhizosphere microorganisms, the diversity of grass microorganisms increased, and the diversity of legume microorganisms decreased (Figure S2). The dominant phylum of fungi detected in the rhizosphere and bulk soils were *Ascomycota, Basidiomycota, Zygomycota, Glomeromycota, Chytridiomycota*, which is consistent with findings regarding the application of organic fertilizers to other agricultural soils [2,42]. Our results showed that there were significant differences in the rhizosphere microbial community structure of grass and legumes after the application of OF compared with non-OF, and LEfSe found that the *Entrophosporaceae, Entorphospora*, and *Entrophosporaneradensis* in the grass rhizosphere (MG) community increased significantly after the application of OF (Figure S3). Arbuscular mycorrhizal fungi (AMF) can form mutually beneficial symbiotic relationships with the roots of more than 80% of terrestrial plant species (about 200,000 species). The diversity of soil AMF has been studied in different agricultural systems, and it was found that their diversity in organic farmland was significantly higher than that in conventional farmland [43]. *Entrophosporaceae, Entorphospora*, and *Entrophosporaneradensis* are newly discovered species of AMF [44], and they were found to be significantly increased in the rhizosphere of grass after the application of OF. Numerous studies have confirmed that AMF can enhance host plant resistance to drought, salt, heavy metals, extreme temperatures (high and low temperatures), acidic soils (low pH), aluminum toxicity, and other pollutants (arsenic contamination and PAHs) [45–47]. The potential mechanisms of AMF to improve the tolerance of host plants to various abiotic stresses include enhancing host nutrient uptake, improving water utilization, enhancing photosynthesis, and scavenging reactive oxygen species (ROS) [44,48]. Generally speaking, the diversity of AM fungi determines the functional diversity of farmland ecosystems. The higher the diversity of AM fungi, the better the quality and yield of crops [49].

In ML, significant increases were found in *Fusariumacuminatum, Fusariumtorulosum*, and *Fusariumlateritium*, which are soil-borne pathogens that causes fusarium wilt in crops such as watermelon and barley [50], and may be a potential explanation for why forage biomass did not increase. In addition, the application of OF significantly increased the relative abundance of pathogenic microorganisms in the grassland [51]. LEfSe results showed that *Fusariumtricinctum, Phomaglomerata*, and *Microbotryomycete* significantly increased after OF application in the bulk soil fungal community. It has been reported that *Fusariumtricinctum* can affect the yield and quality of *Triticum aestivum* L. and also infect different crops [52]. *Phomaglomerata* is distributed in seeds, soil and plant residues and can cause leaf spot and soft rot. *Phomaglomerata* has a wide host range and can affect nearly one hundred plant species. The increase in *Phomaglomerata* will cause inestimable losses to the ecological environment and agriculture [53,54]. *Microbotryomycete* is an important plant pathogen, which can cause serious diseases in crops and forage [55,56]. In conclusion, the increase in the relative abundance of these pathogens may be the reason for the unchanged forage biomass.

The application of OF is an important measure to improve soil quality and promote plant growth in agricultural systems [57,58]. Previous reports mainly studied the response of organic fertilizer application to soil microbial composition and structure [59,60]. Here, network analysis evaluated the structural differences of soil microbial after applying

OF to the planting patterns of grass–legume mixtures, thus providing new insights into the underlying mechanisms of OF treatments. The constructed network map revealed that the application of organic fertilizer reduced the network size of the rhizosphere fungal community, while the network size of the prokaryote community did not change significantly (Figure 4, Table 2). Some studies have indicated that the OF can lead to the formation of potential functional groups and the interaction of microorganisms. Compared with bacteria, fungi are more sensitive to exogenous OF [61]. Before the application of OF, the microbes in the networks of CKMG (99.1%) and CKML (95.9%) were all positively correlated, but after the application of OF, the positive correlation of microbes in the ML network was reduced to 83.7%, while positive correlations still dominated the MG network. Previous studies have shown that the nutrients of OF are more selective on fungi, thereby enhancing the impact of niche filtering on the fungal communities. Therefore, the application of OF will promote species reduction, thereby reducing fungal abundance [62]. The mutual competition between fungi in the ML may restrain the formation of complex network modules, which may also be a cause for the difference in MG and ML production after OF application for plant-growth-promoting rhizobacteria.

Microbial interactions enhance the complexity and stability of soil microbial community structure, which is important for soil health and plant growth [63,64]. In addition to the complexity of the network, microbial stability is also critical to ecological processes [65]. In our study, the number of network nodes and edges in the fungal and prokaryote communities of the grass–legume mixtures were reduced, which indicated that the robustness of the network stability was not significantly different from that without OF; however, the vulnerability was higher than without OF (Figure 4a). Studies have shown that the application of OF can reduce symbiosis and affect interactions between fungi [51], which indicates that the application of OF led to a more unstable fungal network. In the prokaryote network structure, the robustness of the legume network without OF was significantly higher than that of the one treated with OF, and there was no significant difference for the grass and bulk soil prokaryote communities. Meanwhile, compared with the non-OF, the network vulnerability of the OF was higher (Figure 4b). Our results are similar to previous findings that the application of organic fertilizer may reduce microbial interactions [66] and, thus, be detrimental to soil health and indirectly result in the lack of increased grass production.

Soil microorganisms can affect crop biomass directly through the recruitment of beneficial bacteria by plant roots to form symbiotic relationships, or indirectly affect crop biomass by changing soil nutrients [63,67]. Our SEM analysis showed that the application of OF did indeed affect the biomass of forage by affecting soil physicochemical properties, such as TN and TOC, rhizosphere and bulk soil microorganisms, and microbial network structure (Figure 5). Our results showed that there was a significant negative correlation between the application of OF and soil physicochemical properties, and the stability of the fungal community network structure was significantly negatively correlated with forage biomass. There was a significant negative correlation between the application of OF and the stability of the prokaryote community network, and a significant positive correlation between the stability of the prokaryote community and forage biomass, which was consistent with our previous results, which suggests that prokaryote and fungal communities greatly impact crop yield in agricultural ecosystems [5,68,69]. Therefore, our results further proved that the application of OF only changed the physical and chemical properties of the soil in the grass–legume mixtures, but did not increase the forage biomass by affecting the soil microbial diversity, network complexity, and stability.

## 5. Conclusions

In conclusion, this experiment with grass–legume mixtures in the alpine grassland of the Qinghai–Tibet Plateau showed that OF application can indeed ameliorate the physical and chemical properties of soil, but there was no further promotive effect on the forage biomass due to changes in the microbial community. Network analysis showed that the application of OF significantly reduced the network size of the rhizosphere fungal

communities of grass and legumes. Co-occurrence network analysis further indicated that the fungal community network was not stable enough under the OF treatment, which may reduce soil microbial interactions and thus affect forage biomass. Moreover, SEM analysis further showed that OF was negatively correlated ($-0.357$ *) with soil physicochemical properties and stability of the prokaryote community network, and the stability of the fungal community network was negatively correlated ($-0.702$ ***) with the forage biomass. Thus, the OF application had no significant effect on the biomass of forage, because this application did not promote the interactions between rhizospheric microbes and forages. Our experiment may be limited by its short experimental duration. However, it still provides a new understanding of the effects of OF treatment on forage biomass for grass–legume mixtures and provides a theoretical basis for regulating the microflora of grass agronomy.

**Supplementary Materials:** The following supporting information can be downloaded at: https://www.mdpi.com/article/10.3390/agronomy13020481/s1, Figure S1: Diversity of microbial communities in the OF and non-OF bulk soil and rhizosphere soil based on 16s rRNA gene and ITS gene sequencing. (a) Monoculture; (b) Grass-legume mixtures. * indicates significant differences ($p < 0.05$). Under the monoculture: G: rhizosphere samples of OF; CKG: rhizosphere samples of non-OF; BG: bulk soil samples of OF; CKBG: bulk soil samples of non-OF. Under the grass-legume mixture: MG: rhizosphere samples of grass OF; CKMG: rhizosphere samples of grass non-OF; ML: rhizosphere samples of legume OF; CKML: rhizosphere samples of legume non-OF; BF: bulk soils samples of OF; BCK: bulk soil samples of non-OF. Figure S2: NMDS plot of rhizosphere and bulk soil microbial communities in organic fertilization and non-fertilization under grass-legume mixtures. Note: MG: black squares represent grass rhizosphere OF; CKMG: green circle represents non-OF in the rhizosphere of Grass; ML: pink triangle represents legume rhizosphere OF; CKML: blue pentagram represents the legume rhi-zosphere non-OF; BF: yellow diamonds represent bulk soil fertilization; BCK: blue triangle represents bulk soil non-fertilization. Figure S3: Microbial composition of rhizosphere and bulk soils under grass-legume mixtures. (a) Fungi, (b) Prokaryotic. MG: rhizosphere samples of grass OF; CKMG: rhizosphere samples of grass non-OF; ML: rhizosphere samples of legume OF; CKML: rhizo-sphere samples of legume non-OF; BF: bulk soils samples of OF; BCK: bulk soil samples of non-OF. Figure S4: Histogram of LDA SCORE (log 10) distribution. (a) Comparison of Grass (MG) and legume (ML) in rhizosphere fungal com-munities. (b) Comparison of bulk soil (BCK V.S. BF) for bacterial and fungal communities. Table S1: The treatments and the number of replicates in this experiment. Table S2: Physicochemical properties and forage biomass of soils from grass-legume mixtures under organic fertilization and non-fertilized. Under the grass-legume mixtures, MG: rhi-zosphere samples of grass OF; CKMG: rhizosphere samples of grass non-OF; ML: rhizosphere samples of legume OF; CKML: rhizosphere samples of legume non-OF; BF: bulk soils samples of OF; BCK: bulk soil samples of non-OF.

**Author Contributions:** Conceptualization, Y.D.; Methodology, H.Y. (Huilin Yan), S.G., H.Y. (Hao Yu), Y.W. and H.Z.; Formal analysis, H.Y. (Huilin Yan), K.M. and Y.Z.; Investigation, Y.D.; Resources, X.Z., K.Z., H.L. and D.S.; Writing—original draft, H.Y. (Huilin Yan); Supervision, G.L.; Project administration, Y.D. All authors have read and agreed to the published version of the manuscript.

**Funding:** This work was supported by the International Cooperation Project of Key Research and the Development and Transformation of Science and Technology Department of Qinghai Province (2023-HZ-803); the Key R&D and Transformation Projects in Qinghai Province (2022-SF-147); the National Key Research and Development Program of China (2022YFE0114000); and the Fundamental Research Project of the Qinghai Provincial Department of Science and Technology (2021-ZJ-915).

**Data Availability Statement:** Data has been deposited in the China National Microbiology Data Center (NMDC) with the accession number NMDC10018278.

**Conflicts of Interest:** The authors declare that there are no conflicts of interest regarding the publication of this paper.

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
