# Peer review of "Response of Organic Fertilizer Application to Soil Microorganisms and Forage Biomass in Grass–Legume Mixtures"

_agronomy, doi:10.3390/agronomy13020481_

Round 1

Reviewer 1 Report

Specific comments

As a researcher in the field of farming, I am very interested in your work. I have looked thoroughly at your article and I see that you did a lot of work on it.

However, There are some problems in the article that need to be solved, if I understand your description correctly. As far as I see, the paper can be accepted if the points below are dealt with appropriately.

Abstract

1. L15-32: Data is missing in the abstract, so data support should be added.

2. It is suggested that the research content and purpose of this experiment be expressed simply and clearly. The results are more expressed, and some can be removed.

3. Insufficient clarity and adequacy of the purpose of the study in the abstract.

4. No description of the processing in the abstract, making the results difficult to understand.

5. The conclusion section of the abstract is too lengthy.

6. Lack of description of experimental treatments in the abstract.

Keywords

7. Suggested addition of soil microorganisms to keywords.

Introduction

8.  L79: Lack of references in the article.

9.  L85-89: Lack of references in the article.

10.  L49: Medicago sativa L. (legume) "L" should not be italicized.

11. Reduce abbreviations in the full text and increase the readability of the article.

12.  L94-107: Lack of references in the article.

13. A brief introduction to soil microorganisms.

14. L62-64, 77-79, it is suggested to reduce some words like "... less research" in the article.

15. At the end of the introduction, it is suggested to state the purpose of the study.

16. There is a lack of information about the research on forage and crop mixes.

17. L99: Information on the species of the study material should be placed in Materials and Methods.

18. L103–107: The conclusion should not be addressed in the introduction, but rather the assumptions made here. Page 1-2, line 36-93. The introduction part is loose in structure, so it is suggested to condense the language.

19. L103-107. It is the first time to see a conclusive conclusion in the introduction part. Generally, this part is the research content or hypothesis, and it is suggested to reconsider.

20. L47, lack of introduction on the advantages of mixing forage with crops.

21. L49-51, please give a brief description of these forage species.

22. L103, with a specific conclusion in the introduction, this experiment would have no practical significance. Please state the hypothesis in the introduction.

Materials and Methods

23.  L109: Please check the full-text format and whether the numbers are in the new Roman format.

24.  L123-124: Do you need spaces between English and numbers?

25.  L129: Add a space between the line number and the unit.

L115: 12 was Beilin201 and Chuancao; 12 was Beilin and Aba” missing space before 201, add a space before 201.

26.  L119: The treatment conditions were set as non-OF and 50kg/acre of OF, Spaces should be added between numbers and units.

27.  L130: A quadrat of 50cm×50cm was selected for sampling to collect rhizosphere and bulk soil. Spaces should be added between numbers and units.

28.  L124: Beilin201 was seeded at 45 g/15 m2. Check whether the unit is correct.

29.  L160: “57°C” Add spaces between data and units.

30.  L148: “DNA extraction, amplification, sequencing, and sequence analysis”. Check whether the format in the article is correct.

31.  L180: “using the Unoise3 algorithm (which removed 180 OTUs with abundances of less than 8 sequences” Add a space before the number.

32.  L219: (http://mem.rcees.ac.cn:8081) Link format error.

33. 2.1 In case of more content, it can be divided into two paragraphs to explain or add a title.

34. L109, the description of the experimental site is less and very sloppy.

35. L126-127, only random sampling was carried out and without repetition.

36. L 118-119, “-” is this symbol correct?

37. L 152/164, “~” is this symbol correct?

38. In which year was the experiment conducted? And when is the sowing time and harvest time?

39. The basic physicochemical properties of the soil in the test site were not described.

40. How many repetitions of the experiment are there?

41.  No mention of irrigation was made in the experimental design.

42. The article has more treatments in the experimental design and the description of the ideas is not clear enough, which makes each treatment difficult to understand.

43. L130–132: “At the seedling stage of herbage growth, legume and grass forage were similar in root length and diameter, so we collected 15 cm of soil, including the roots of forage” is missing a reference, and the “15 cm” here refers to the depth or width needs to be clear.

44. L133–134: What is the method used for root separation?

45. L147: Is it table 1 or table S1? Table S1 not found.

46. L 219-220,it is best to replace the word random with stochastic,as it is a contraction of RMT,which an be confusing.

47. L111, the year in which the experiment was carried out is not described, please add it.

48. L111, no description of the physical properties underlying the experimental site.

49. L114, no specific experimental treatments and number of replications for monoculture and mixed crops are presented.

50. L114, which does not describe the specific manner and frequency of irrigation at the experimental site.

51. L121, where the introduction of mixed seeding treatment is unclear, suggests reorganising the language.

52. L130, please add the method of collecting the soil sample.

53. L132, the experimental width and depth of the experiments are not clearly described.

54. L146, using composite samples is inaccurate for the number of experimental replicates.

55. L147, this article does not have a table for Table S1, please add it.

Results

56. Lines 247-248,Forage biomass under different planting systems. (a) Plant height; (b) Plant fresh weight; (c) Plant dry weight. ns means that there is no significant difference between the groups . This sentence seriously inconsistent with Figure 1.It is recommended to modify the note or data graph.

57. L258 and 263,P>0.05should be changed P<0.05.

58. L351-352,the notes in Figure 3 is suggested serious consideration for modification.

59. L433,TN, NO3-N, NH4-N, TOCshould be changed TN, NO3-N, NH4-N, TOC.

60. L452-453,*** and ** represent significance at p<0.01 and p<0.05 respectively). should be changed *** and ** represent significance at P<0.001 and P<0.01 respectively). .

61. This part is too verbose, it is recommended to simplify the language.

62. L109. It is suggested that this part be described separately for reading.

63. There is no need to elaborate too much on common sense.

64. L235, Table S2 is not available in this article, please add it.

65. L247, Figure 1 is missing the titles of the horizontal and vertical coordinates, also the descriptions of a, b and c are missing from the figure, please add them.

66. L263 and 266, Figures S1, S2, S3 and S4 are missing to confirm the accuracy of the data, please add them.

67. L290, the number of groups in Table 2 for 16S and ITS does not match, please explain.

68. L351, there is no description of a and b in Figure 3, please add.

69. L434, "98% of the forage biomass (plant fresh weight and plant dry weight)" does not match the data in Figure 5.

70. It is suggested that an explanation of the abbreviations be included in the illustrations to make them easier to read.

71. Lack of analysis of soil physicochemical properties in the analysis of results.

72.  L247: a, b, c” is missing from the figure.

73.  L285: There is one more space in the 16s r RNA in Table 1.

74.  L411: “but not significantly so”. Check for errors.

75.  Please check whether the title format in the full text is correct.

76. The figure 1 does not correspond to the annotation.

77. Table 1, there are many decimal places.

78. L391-398, please explain *** and ns.

79. L247: Plant height is not found in Figure 1.

80. L263, 266: Figure S1 and S2 not found.

81. The black and gray arrows in Figure 5 are incorrectly described in the legend, please revise the legend.

82. L358–366: The “avg K” value does not match the data in Table 2.

83. Line 247-248. Plant height? What does CKG and CKMG and MG mean? Please mark clearly.

84. L286-290. Please place this part below Table 1.

85. Please note the details in Table 1, where is the monoculture in the ITS gene Rhizosphere?

86. L325-330. There are differences between other color groups, so is it necessary to distinguish between significant and extremely significant?

87. Please write R2 correctly in Table 2.

88. L434. 98% of the biomass? Please check the data against Figure 5.

Discussion

89. The yield section is missing from the discussion.

90. The discussion did not cover the relevant literature on the effects of organic fertilizer application on soil physicochemical properties and soil microbial diversity, etc.

91. Extreme weather occurs frequently on the Qinghai-Tibet Plateau. Did extreme weather occur in the year in which the test was conducted and did it affect the test, and if it was a one-year test, were the data representative?

92. L475,Consistent with previous results”,there is a lack of references.

93. L484-485,and application OF did not increase the grass biomass in grass-legume mixtures, compared with non-OF.This conclusion is a little hasty, it can explore the relationship between soil characteristics and biomass, and finally come to a conclusion.

94. The comparison of data under mono-cropping and intercropping conditions is missing from the discussion.

95. It is suggested that this part be listed separately for reading.

Conclusion

96. Lack of data in the conclusion

97. In the final conclusion, limitations are not discussed and state the research focus and prospect of the follow-up work.

98. L 615-616 ,that OF treatment prevented formation of a stable network,is suggested that the fungal community network is not stable enough under the OF treatment.

99. L 475, please add relevant references to verify the experimental results.

100. L 483, this article does not make any assumptions.

101. L500, please explain the reason for this result.

102. L508, 518, and 524, too much use of "our study showed", please change the wording.

103. L580, where the discussion should focus on the stability of the soil microorganisms rather than the stability of the network.

104. L 604, most of the results of this article are consistent with the previous findings of the authors' team, so what is the innovation of this article?

References

105. The format of references cited in the full text is incorrect.

106. References do not need to add numbers.

107. Why do all documents have duplicate numbers?

108. L219, there is a format error in the article []. It is recommended to check the full text.

109. There are numerous problems with the format of the references, such as wrong symbols in the authors, italicized issue numbers and journal names, etc.

110. L643, Annual N2O emissions from conventionally should be changed Annual N2O emissions from conventionally.

111. There are also many problems in the reference, such as repeated serial numbers, inconsisitent case and missing page numbers.

112. Please verify the name format and add DOI.

Reviewer 2 Report

Really interesting paper on the effects of organic and non-organic fertiliser applications on soil microbial communities in a grass-legume system. Overall, the paper is well written and structured. I have only a few minor comments.

1.       Labelling of the variates is confusing – I kept having to refer back to the methods to remind myself what CKMG, MG, ML etc referred to, which interrupted the sense of the text. It would be much easier for the reader if the variables (rhizosphere, bulk soil etc) could either be written in full so it’s obvious in the text what is being referred to, or else acronyms chosen which more obviously relate to their full English name.

2.       Figure 1 legend doesn’t seem to relate to the figure that is shown and the y axis isn’t properly labelled

3.       I can’t comment on the genetic analysis, but the first para of section 3.2 looks more like it belongs in the methods section. Same for fist sentence of next para

4.       Figure 3 legend not clear and (a) and (b) aren’t labelled in the figure

5.       Figure 4. Legend says “with or without OF amendment” but which is “with” and which is “without” – these aren’t labelled on the figure itself, though maybe inferred by the coding system which isn’t obvious

Final general comment – the discussion section is interesting and well written, but I wonder whether the authors could comment on the temporal aspect of their findings? Are the results due to an initial disturbance or shock to the system from adding in OF? And would this be likely to stabilise over time resulting in a more positive impact on plant growth?

Reviewer 3 Report

Manuscript title “Response of organic fertilizer application to soil microorganisms and forage biomass under Grass-legume mixtures” is well-written manuscript with extensive work.

Paper will be very interesting for the reader of ecologist and Microbiologist. I have few points to be clear in the paper, and those should be clarified before the paper would be accepted.

Abstract

1.     Abstract not consist any mechanisms or concept involve in such findings

2.     Abstract should have crisp information about the aim conclusion, which I don‘t find in the present abstract form.

3.     It always crucial to consider ecological importance also of the proposed manuscript which I find a missing link.

4.     Microbial diversity (Bacterial and Fungal) need to be write more clearly in relation to OF.

5.     Which kind of stains are highly affected and which one is more dominating?

6.      How such studies will help the ecologist to deal with Environmental and Ecological issues?

7.      In case of Rhizosphere various PGPRs is there any findings on these strains?

Round 2

Reviewer 1 Report

Thanks for revising the paper. I have checked the revision and I am satisfied with the changes you made to this manuscript.